# Differential Allele-Specific Expression Revealed Functional Variants and Candidate Genes Related to Meat Quality Traits in *B. indicus* Muscle

**DOI:** 10.3390/genes13122336

**Published:** 2022-12-11

**Authors:** Jennifer Jessica Bruscadin, Tainã Figueiredo Cardoso, Wellison Jarles da Silva Diniz, Marcela Maria de Souza, Juliana Afonso, Dielson Vieira, Jessica Malheiros, Bruno Gabriel Nascimento Andrade, Juliana Petrini, José Bento Sterman Ferraz, Adhemar Zerlotini, Gerson Barreto Mourão, Luiz Lehmann Coutinho, Luciana Correia de Almeida Regitano

**Affiliations:** 1Center of Biological Sciences and Health, Federal University of São Carlos, São Carlos 13560-000, SP, Brazil; 2Embrapa Pecuária Sudeste, São Carlos 13560-000, SP, Brazil; 3Department of Animal Sciences, Auburn University, Auburn, AL 36882, USA; 4Department of Animal Science, Iowa State University, Ames, IA 50011, USA; 5Department of Internal Medicine, Yale University School of Medicine, New Haven, CT 06520, USA; 6Federal University of Latin American Integration-UNILA, Foz do Iguaçu 85851-000, PR, Brazil; 7Computer Science Department, Munster Technological University, MTU/ADAPT, T12 P928 Cork, Ireland; 8Center for Functional Genomics, Department of Animal Science, 13400-000, University of São Paulo (ESALQ—USP), Piracicaba 13400-000, SP, Brazil; 9Department of Veterinary Medicine, University of São Paulo (FMVZ—USP), Pirassununga 13630-000, SP, Brazil; 10Embrapa Informática Agropecuária, Campinas 13010-000, SP, Brazil

**Keywords:** phenotype, beef, *cis*-regulation, ASE

## Abstract

Traditional transcriptomics approaches have been used to identify candidate genes affecting economically important livestock traits. Regulatory variants affecting these traits, however, remain under covered. Genomic regions showing allele-specific expression (ASE) are under the effect of *cis*-regulatory variants, being useful for improving the accuracy of genomic selection models. Taking advantage of the better of these two methods, we investigated single nucleotide polymorphisms (SNPs) in regions showing differential ASE (DASE SNPs) between contrasting groups for beef quality traits. For these analyses, we used RNA sequencing data, imputed genotypes and genomic estimated breeding values of muscle-related traits from 190 Nelore (*Bos indicus*) steers. We selected 40 contrasting unrelated samples for the analysis (N = 20 animals per contrasting group) and used a beta-binomial model to identify ASE SNPs in only one group (i.e., DASE SNPs). We found 1479 DASE SNPs (FDR ≤ 0.05) associated with 55 beef-quality traits. Most DASE genes were involved with tenderness and muscle homeostasis, presenting a co-expression module enriched for the protein ubiquitination process. The results overlapped with epigenetics and phenotype-associated data, suggesting that DASE SNPs are potentially linked to *cis*-regulatory variants affecting simultaneously the transcription and phenotype through chromatin state modulation.

## 1. Introduction

Allele-specific expression (ASE) analysis is a helpful approach to searching for *cis*-regulatory variants [1,2]. ASE consists of the imbalanced ratio between reference and alternative alleles’ counts in a heterozygous locus, being a common pattern in mammals [3,4,5]. As ASE is determined per individual, it reduces the effect of technical factors found in total expression studies [1,6]. Thus, from the knowledge of the transcribed or repressed alleles it is possible to infer *cis*-regulatory variants related to this expression pattern [1], allowing increased accuracy in predictive models and improving functional variants discovery [7].

As our research group [8,9] and others reported, ASE is widespread in bovine [4,6,8,10,11]. Previously, we described a high coverage list of single nucleotide polymorphisms (SNPs) located in transcripts with ASE (ASE SNPs) distributed in Nelore (*Bos indicus*) muscle and the candidate *cis*-regulatory variants potentially affecting the allelic expression. Additionally, we have shown evidence that the ASE SNPs can regulate transcription, as they are located in functional regions [9]. ASE analysis benefits the search for functional variants affecting important economic traits, since genes containing ASE SNPs (ASE genes) may play a role in muscle maintenance and are related to meat quality traits [8,10].

A traditional transcriptomics analysis compares the total gene expression between phenotypically contrasting animals [12], resulting in differentially expressed genes (DEGs). Other techniques are needed to investigate variants associated with the DEGs expression (expression quantitative trait loci—eQTLs, for example) and associated with the phenotype (quantitative trait loci—QTLs). Thus, these data layers need to be integrated to draft more consistent potential mechanisms. Facing the benefits of conducting differential expression and ASE analyses, a differential allele-specific expression (DASE) approach can identify regions where the ASE level differs between extreme samples for a given trait.

The DASE regions are likely affected by *cis*-regulatory variants affecting the expression pattern because of the observed ASE. In addition, the same *cis*-regulatory variants may be related to the phenotype of interest due to allelic imbalance being trait-dependent. Thus, this approach can reduce analytic steps and avoid false positives compared to integrating results from multiple consecutive analyses. There are only a few studies with global DASE analysis, probably due to the requirement of high density genotypic, transcriptomic and phenotypic data from the same sample. Genome-wide DASE regions were related to breast cancer phenotypes [7,13], but no study could be found in livestock so far.

In this study, we searched for DASE SNPs related to 55 meat quality, carcass quality, mineral content, and fatty acid composition traits in the muscle of a Nelore population. We conducted gene enrichment and co-expression analyses to investigate functional interactions between DASE genes. We integrated the DASE SNPs with results obtained previously in the same population, such as *cis*-eQTLs (i.e., SNPs associated with the transcript abundance) and aseQTLs (i.e., SNPs associated with the allelic imbalance). These variants may affect the traits by the cumulative regulatory effect in co-expressed genes related to these muscle-related traits.

## 2. Materials and Methods

### 2.1. Animal Production and Sample Collection

The experimental population used in this study was part of a large project with all experimental approaches approved by the Institutional Animal Care and Use Committee Guidelines of Embrapa Pecuária Sudeste ethics committee (São Carlos, São Paulo, Brazil. Protocol CEUA 01/2013). Details regarding the animal’s production can be found inprevious studies [14,15,16]. For the animal production, 34 sires were selected to have the lesser kinship and to represent the main lineages of Nelore cattle in Brazil. Semen samples of these animals were stored at 80 °C for later analysis. Artificial inseminations were performed in purebred Nelore dams, avoiding the same parental pair [15], generating a 386 Nelore steers population [16]. The animals were raised under the same feeding and management as detailed described previously [15]. The animals were slaughtered once they reached 5 mm of backfat thickness (BFT), measured by ultrasound. Blood samples and collected in vacuum tubes containing potassium EDTA (K3) and muscle samples were immediately collected between the 11th and 13th ribs of the *Longissimus thoracis* muscle [14,15,16]. Here, we used the data of 190 steers from this experimental population for which phenotype, genotype, and RNA-Seq data were available, being the same samples used in previous studies of our group [8,9], to ensure comparability.

### 2.2. Phenotype Measurements

Phenotypes’ description and genomic estimated breeding values (GEBVs)’ summary are described in Appendix A. Tenderness, Backfat Thickness (BFT), and Ribeye Area (REA) traits were collected as described previously [15] Briefly, tenderness phenotypes were measured based on the shear force obtained from the Warner-Bratzler equipment at 24 h (WBSF0), seven days (WBSF7), and 14 days after slaughter (WBSF14) [15]. The phenotypes for mineral content (Ar, Ca, Cr, Co, Cu, Fe, Mg, Mn, P, K, Se, Na, S, and Zn) were obtained from 100 mg lyophilized muscle sample, being measured by a Vista Pro-CCD ICP-OES spectrometer with a radial view (Varian, Mulgrave, Australia) [17]. GEBVs for these phenotypes were calculated in GenSel software (https://github.com/austin-putz/GenSel, accessed on 10 April 2020), using birth, feedlot locations, and breeding seasons as fixed effects, and age at slaughter was included as a covariate in the statistical model [15,17].

The intramuscular fat (IMF) and fatty acid composition traits (described in Appendix A) were previously reported elsewhere [16]. The measures were conducted in the Animal Nutrition and Growth Laboratory at ESALQ, Piracicaba, São Paulo, Brazil. IMF was obtained using an Ankom XT20 Fat Analyzer, following the recommended methods. Fatty acid content was extracted from 4 g of muscle samples and the profile identification was performed using gas chromatography by comparing the retention time of the obtained methyl esters with reference standards. Detailed methods are described previously [16], which also presented the model for the GEBV estimation for these traits.

The selection of animals to compose contrasting sample groups was similar to the criteria adopted in the previously reported differential expression data [18,19,20,21,22]. The animals were ordered and grouped according to the GEBV values for each trait, being the group with higher GEBV values classified as High and the one with minor values classified as Low. The animals were chosen to avoid steers descending from the same sires being allocated to the same contrasting group. A Student’s *T*-test was applied for all mentioned traits (*p*-value < 0.05) to determine whether the phenotypes differed between the two groups.

To observe the relationship between phenotypes in our dataset, we computed the Pearson correlation between GEBVs values from the subpopulation of 190 animals in a correlogram.

### 2.3. DNA Extraction, Genotyping and Whole Genome Sequencing

DNA was extracted from frozen semen samples of the sires using the phenol-chloroform method. Extracted DNA from 26 sires was used for whole-genome sequencing, performed in an equipment Illumina HiSeq 2500 System (Illumina Inc., San Diego, CA, USA), with 8–21X coverage, in the Functional Genomic Center—ESALQ (Piracicaba, SP, Brazil). Data management was executed as suggested by the 1000 Bulls Genomes Project (http://www.1000bullgenomes.com/, accessed on 1 July 2020), as previously described [23].

Concerning the progeny, DNA was extracted from blood samples using the salting out method, as described previously [14]. For sires and steers samples, the DNA concentration was quantified in NanoDrop^®^, purity was analyzed by comparing the optic absorbance ratio 260/280, and the material integrity was observed by electrophoresis in agarose gel. Genotyping was conducted in Illumina BovineHD BeadChip (Illumina Inc, San Diego, CA, USA) with the extracted DNA from sires and steers, as detailed described elsewhere [24].

Variant imputation was executed with the high-density genotypes together with the 26 sires’ whole-genome sequencing data, which methods were described elsewhere [23]. Concisely, the imputation accuracy was analyzed using a leave-one-out method, where the sequenced genotypes were used as a reference to observe consistent imputation results. Quality control was conducted with PLINK software [25], maintaining autosomal SNPs with minor allele frequency higher than 0.01, which showed an imputation error rate of less than 2% [23]. The final genotype file contained nearly 4.8 million SNPs.

### 2.4. RNA Extraction and Sequencing

RNA from 100 mg of 190 muscle samples from steers was extracted for transcriptomics analysis using TRIzol^®^ (Life Technologies, Carlsbad, CA, USA), according to the manufacturer’s instructions. RNA libraries were prepared with the TruSeq RNA Sample Preparation Kit (Illumina, San Diego, CA, USA) with the default protocol. Clustering and paired-end sequencing were executed in an Illumina HiSeq 2500^®^ (Illumina, San Diego, CA, USA) equipment. The quality control removed sequencing adapters and low complexity reads with the software SeqyClean *v*1.4.13 [26]. The RNA sequencing was made in the Functional Genomic Center—ESALQ (Piracicaba, SP, Brazil), being detailed in a previous manuscript [27].

The sequences were mapped to the bovine reference genome (ARS-UCD1.2, Ensembl 100) using the STAR software [28] with default settings. The WASP pipeline [29] was applied to remove reads presenting mapping bias, which commonly inflates the reference allele counts [30]. Counts of reads overlapping gene regions were obtained using the Htseq-count tool [31], and allele-specific counts were obtained from heterozygous variants with GATK ASE Read Counter [32].

### 2.5. Analysis of Differential Allele-Specific Expression (DASE) between Contrasting Groups for Meat Quality Phenotypes

The DASE analysis was conducted to evaluate the relationship between the allelic expression imbalance level and the phenotype by comparing allelic ratios between contrasting groups. In this approach, tests were performed per phenotype for all SNPs within their respective genes aiming to compare results with those from differential expression analysis, co-expression, and gene functions. We used the DESeq2 R-package [33], including each sample twice in the matrix of counts (for reference and alternate alleles expression), being the design formula defined as:design = ~phenotype + phenotype:sample + phenotype:count
where: ~phenotype corresponds to the variation between High and Low groups for the meat quality traits; phenotype:sample calculates the phenotypic variation of the animals within each group; and phenotype:count estimates the count’s difference between reference and alternate alleles for the two contrasting groups. Unlike the traditional differential expression analysis, size factor normalization is unnecessary in this approach because the sample is included in the design formula. The code and model details can be found on the tutorial page, described by the DESeq2 developer (http://rstudio-pubs-static.s3.amazonaws.com/275642:e9d578fe1f7a404aad0553f52236c0a4.html, accessed on 12 September 2022). In this analysis, each test results in three *p*-values. One determines the significance of the difference between allele counts within the High group, other for the Low group, and the last *p*-value corresponds to the ASE significance between the two contrasting groups, which is adjusted for the False Discovery Rate (FDR) implemented by the Benjamini-Hochberg method. The script also returns Log2FoldChange (FC) values, indicating which contrasting group showed allelic imbalance. The negative FC values indicate that the allelic imbalance was larger in the Low group compared to the High group. The positive ones correspond to variants in which the ASE was downregulated in the Low group compared to the High group. This analysis resulted in DASE SNPs, which corresponded to the SNPs marking the significant DASE within a gene, which were named DASE genes. We considered as significant the DASE SNPs corrected per phenotype and per gene with FDR ≤ 0.05.

### 2.6. Functional Annotation and Gene Enrichment Analyses

The DASE SNPs were annotated using SNPEff software [34] against *B. taurus* genome coordinates. Gene ontology (GO) terms related to biological processes and molecular functions involving DASE genes were obtained from BiomaRT (Ensemble release 104). The enriched KEGG metabolic pathways containing DASE genes were obtained with ClueGo Plugin for CytosCape [35], release 101, with the two-sided hypergeometric test in the enrichment analysis (FDR ≤ 0.05).

### 2.7. Co-Expression Analysis

To evaluate the correlation between DASE genes, we performed a co-expression analysis using the default parameters of the CEMITool package [36]. This software results in outputs with co-expressed modules and performs gene set enrichment (GSEA) and over-representation (ORA) analyses. The GO biological processes terms underlying the DASE genes were used for ORA (FDR ≤ 0.1). In addition, the phenotype with the largest number of DASE genes was chosen for GSEA. The animals for this trait were separated as High, Low, or “others” according to their GEBVs, “others” being the remaining animals excluding those belonging to contrasting groups. This analysis was conducted out to investigate the relevance of co-expressed genes in each module.

### 2.8. Data Integration

We conducted data integration analysis to examine if the SNPs found in this work were previously related to ASE and bovine phenotypes in other studies. First, we retrieved ASE data from the same Nelore population, regardless of the phenotypes [8,9]. After that, to compare the potential effect on the phenotypes, we downloaded all GWAS data obtained in the muscle of Nelore from previously published manuscripts related to meat quality [15], fatty acid content [16], and mineral concentration traits [37]. QTLs coordinates obtained from the previous bovine reference genome version (UMD3.1) were converted to the latest version (ARS-UCD1.2) using the liftOver tool from the UCSC database (https://genome.ucsc.edu/cgi-bin/hgLiftOver, accessed on 7 June 2022).

DASE genes were compared with DEGs from the same Nelore population, identified between contrasting samples for meat quality [18,21], fatty acid composition [19], and mineral content [20,22] traits.

To identify SNPs potentially regulating the expression pattern, we considered a window of 200 kb upstream and downstream of each DASE SNP to search for aseQTLs [9], *cis*-eQTLs [9] and differentially methylated SNPs (DM) [38]. This 200 kb window was chosen based on our previous study [23] where the aseQTLs were mostly located until 200 kb of the regulated ASE SNP, and based on the high LD with the DASE SNPs expected to be present in that genomic distance.

DASE SNPs and their 200 kb-distant potential regulatory SNPs were compared with ChIP-Seq and ATAC-seq data obtained in cattle muscle for the Functional Annotation of Animal Genomes (FAANG) [39]. The significance of all comparisons with genomic position ranges was tested with 1000 permutations performed by the RegioneR package [40], with a *p*-value threshold of 0.05.

## 3. Results

### 3.1. The Muscle-Related Phenotypes Are Correlated

Herein, we evaluated the potential effect of ASE on 55 carcasses and meat quality, mineral content, and fatty acids composition traits obtained in Nelore steers’ muscle. GEBVs and the summary by sample group used here are reported in Appendix A. The average GEBVs differed significantly between the contrasting groups for all phenotypes (*p*-value ≤ 0.05). After selecting animals with extreme phenotypes for the different traits (N = 20 per contrasting groups), 187 from the initial 190 samples were kept for analysis. The animals could be analyzed in more than one phenotype. The average number of phenotypes tested per sample was 11.76, ranging from one to 36 traits.

We used all animals with collected phenotypes and RNA-Seq data (N = 190) to identify correlated trait clusters in our population. From the correlation matrix, we plot the correlogram in Figure 1. We observed two blocks of strong positive correlations, one between the Ca, S, Zn, Na, Mg, and K minerals and the other between the C14:1 *cis*-9, C16:1 *cis*-9, C20:2, C18:3n3, PUFA, C20:5, C20:4, C22:5, and n3 fatty acids, including weak positive correlations with Cr and Cu in the same block. Negative correlation blocks can be observed, being the more consistent involving C10:0, C12:0, C14:0, C15:0, C18:1 trans -10,11,12, C17:0, C18:3n6, C18:0, C16:0 and SFA traits against WBSF0, WBSF7, WBSF14, C18:1 *cis*-13, C18:1 *cis*-12, C18:1 *cis*-11, C17:1, C20:1, C18:1 *cis*-9, and C18:1 total traits. The correlation matrix is presented in Appendix A.

### 3.2. Differential Allelic Imbalance Related to Livestock Traits Was Identified in Bovine Muscle

We used several filters to increase the DASE results’ reliability. A sample could be tested if it exhibited a heterozygous candidate SNP with at least ten counts. Therefore, we set a threshold of at least three animals for which the SNP satisfied our requirements in each contrasting group. Thus, for all SNPs, the sample number that passed these filters was less than the original 20 samples selected to compose the contrasting groups. The average number of samples per tested SNP was 5.88 in the Low and 5.82 in the High groups, ranging from three to 18 animals in both extreme groups (Appendix A).

For the DASE analysis, 14,429 SNPs were analyzed in 387,919 tests, being 1479 significant tests corresponding to 937 unique DASE SNPs (FDR ≤ 0.05). Focusing on each trait, the number of tests ranged from 5871 to 8141, with a mean of 7053 per phenotype. The average of DASE SNPs identified per phenotype was 26.89, ranging from 11 (for the C16:0 trait) to 73 (for WBSF0). All tests and the corresponding significance are represented in Figure 2A and described in Appendix A. Figure 2B represents the significant DASE SNPs number per phenotype.

FCs ranged from −28.00 to 27.97, with an average of −0.41. The three DASE SNPs with the most negative FC values were the rs524639701 (FC = −28.00; trait: C18:C15), the rs441798208 (FC = −27.93; trait: C18:C15), and the rs523288898 (FC = −26.53; trait: Na). The three DASE SNPs with the most positive values were the rs517071070 (FC = 27.97; trait: BFT), the rs520889234 (FC = 25.93; trait: C18:1:C12), and the rs443152461 (FC = 20.49; trait: C22:6). For 742 DASE SNPs the ASE was larger in the High contrasting sample group, and for 737 DASE SNPs this pattern was found in the Low group. Figure 2B represents the number of DASE SNPs according to which phenotype extreme group exhibited ASE.

We used SNPEff to annotate DASE SNPs. We found that DASE SNPs are mostly located in synonymous variation regions (n = 486 occurrences), followed by 3′ UTR regions (n = 318), downstream regions (n = 248), and 152 DASE SNPs were predicted as missense variations (Appendix A).

### 3.3. Genes with Differential Allele-Specific Expression Were Related to Muscle Homeostasis

The 937 unique DASE SNPs are within 656 genes. The average of DASE SNPs per gene was 2.25. The genes showing more DASE SNPs included Heat shock protein family B—small—member 6 (*HSPB6*), ENSBTAG00000052709 (novel), and ENSBTAG00000048585 (novel), with 36, 30, and 22 DASE SNPs, respectively.

To investigate the biological relevance of the DASE genes, we analyzed the gene enrichment of KEGG metabolic pathways, biological processes, and molecular functions (Appendix A). The DASE genes are related to 2246 biological processes, such as regulation of transcription by RNA polymerase II (with 5.64% of the DASE genes), regulation of transcription, DNA-templated (5.48% of the DASE genes), positive regulation of transcription by RNA polymerase II (5.03%), protein phosphorylation (4.11%), positive regulation of transcription, DNA-templated (4.11%), and protein ubiquitination (3.65%) (Appendix A). Regarding molecular functions, 22.25% of DASE genes were identified in protein binding, 16.15% metal ion binding, identical protein binding (10.36%), ATP binding (9.14%), nucleotide binding (8.84%), and other 758 molecular functions (Appendix A).

We used ClueGo to perform the gene enrichment analysis of the KEGG metabolic pathways. Based on the number of genes in each pathway, the largest number was found for pathways in neurodegeneration (26 DASE genes), Alzheimer disease (24), Parkinson disease (21), Prion disease (19), chemical carcinogenesis (19), and diabetic cardiomyopathy (19). Considering the percentage of all genes belonging to the pathways that overlapped our DASE genes, 16.13% of genes from the citrate cycle (TCA cycle) pathway were found in this study, followed by 13.46% of the genes from amino sugar and nucleotide sugar metabolism pathway, and 9.78% from the hypertrophic cardiomyopathy pathway. Figure 3A shows the enriched KEGG pathways from all DASE genes (Appendix A) regarding the number and proportion of DASE genes in the pathways. The interaction of the pathways is represented in the network in Figure 3B.

### 3.4. Co-Expression between DASE Genes

We conducted a co-expression analysis to identify modules of correlated genes that can affect the studied phenotypes. We found five co-expressed modules (M1, M2, M3, M4, and M5). The M1 module gathered 225 DASE genes, with *NFAT5*, *UQCR11*, *UTRN*, *NCOA3*, and *KDM7A* as hub genes; M2 had 98 DASE genes, with the *SMAD5*, *UBE2W*, *OXR1*, *EDEM3*, and *ARID4A* hubs; M3 had 69 DASE genes, with *MMADHC*, *NDUFA5*, *LRRC39*, *FASTKD2*, and *RTN4* hub genes; 60 DASE genes are within the module M4, with *FANCG*, *CCNL2*, *RTEL1*, *STARD3*, and *CXXC1* hubs. M5 had 42 co-expressed DASE genes, being *VIM, ANXA5, SPARC, ANXA2,* and *COL3A1* its hub genes. A total of 145 DASE genes were not correlated. Modules and paired interacting genes are in Appendix A.

To examine the biological relevance of the co-expressed genes, we analyzed the gene ontology biological processes (BPs) of the DASE genes, corresponding to the ORA (Appendix A). This analysis was not significant for M1, M3, M4, and M5. For M2, the protein ubiquitination process was enriched with FDR = 0.07, with ten DASE genes.

Due to the greatest number of DASE genes (n = 56), we selected the WBSF0 trait for GSEA analysis. The GSEA was performed to explore if modules are upregulated or downregulated in response to this phenotype based on the extreme sample groups. WBSF0 showed significantly enriched gene sets for the modules M1 and M2 in the High and Low groups (Figure 4 and Appendix A). These modules presented opposite effects in each sample group, with the M2 module induced in the High group and repressed in the Low group. Similarly, the M1 module showed repression in the High group and induction in the Low group. 

### 3.5. Integration Was Found between DASE SNPs and QTLs for the Related Phenotypes

Integrating GWAS data obtained in the same Nelore population and the DASE SNPs, retrieved 353 overlaps, involving 22 traits and 129 DASE SNPs (Appendix A). The top three QTLs with more overlap with DASE SNPs were REA (n = 86), LFAT (n = 49), and C18:0 (n = 41). Furthermore, the rs134535828 DASE SNP was identified as a *cis*-eQTL in the same population. Likewise, the rs136209194 DASE SNP was previously found as an aseQTL [9] (Appendix A). Four variants were identified as DASE SNPs and QTLs for C18:0, three for REA, two for WBSF7, and two DASE SNPs for WBSF0 were identified as QTLs for WBSF7, as described in Table 1.

DEGs were also found with the identified DASE SNPs. We found 159 overlaps, between 16 traits and 136 genes containing 196 DASE SNPs (Appendix A). Three DASE genes, SAM and SH3 domain containing 1 *(SASH1),* Solute carrier family 25 member 4 *(SLC25A4),* and Diacylglycerol kinase delta *(DGKD),* identified for C18:1 *cis*-9 were differentially expressed for oleic acid content (i.e., the same fatty acid). Kelch-like family member 40 (*KLHL40*) was a DEG for palmitic acid (C16:0) and a DASE gene for the same trait. Striatin Interacting Protein 2 (*STRIP2)* was differentially expressed for REA [21] and has the rs137477165 DASE SNP identified here for REA.

### 3.6. Regulatory Variants Related to the DASE Pattern

We compared DASE SNPs with data from our previous study [9] (Bruscadin et al., 2022). The ASE SNPs were tested as *cis*-eQTLs, and SNPs until 200 kb distants of the ASE SNPs were tested as aseQTLs. We found all DASE SNPs overlapping ASE SNPs from our previous study [9], 23 of them classified as aseQTLs and 47 as *cis*-eQTLs (Appendix A). Since regulatory mechanisms can act in variants neighboring the region with ASE, we searched for potential regulatory variants until 200 kb from the DASE SNPs. We found 9857 aseQTLs and 1274 *cis*-eQTLs in this genomic window, around 302 and 289 DASE SNPs, respectively, from our recent study [9]. De Souza et al. (2022) used a subset of 12 animals from the same population to obtain the methylation profile in the muscle, identifying regions with differentially methylated (DM) SNPs across contrasting groups for the tenderness phenotype. We found 50 DASE SNPs located until 200 kb of distance from these DM regions (Appendix A).

The DASE SNPs and potentially regulatory SNPs until 200 kb around them (DM SNPs, *cis*-eQTLs, and aseQTLs) were compared with FAANG’s ChIP-seq and ATAQ-seq data [39]. We found 182 DASE SNPs overlapping functional regions (Appendix A). From the overlapped DASE SNPs, 21 were previously identified as QTLs in the same population, affecting traits such as REA (15 SNPs), BFAT, C18:0 and LFAT (five SNPs), WBSF7 and WHC (four SNPs). From all potential regulatory variants until 200 kb of distance from the DASE SNPs, 1145 aseQTLs, 212 *cis*-eQTLs and 14 DM SNPs were located within functional regions from FAANG (Appendix A). The permutation tests conducted with RegioneR indicated that the data integration was significantly more frequent than by chance (Appendix A).

We compared all features overlapping DASE SNPs to evaluate the evidence of the regulatory potential on the transcription and phenotype (Appendix A). Some SNPs were found in multiple datasets (Appendix A). The rs519474617, identified as aseQTLs and *cis*-eQTLs [9], is located in a region presenting all epigenetic marks from FAANG’s study obtained in bovine muscle [39]. The rs717173356, found as a *cis*-eQTL and aseQTL in our previous study [9], is located in a CTCF binding region from Kern et al. (2021). The rs470985689 DASE SNP, which showed DASE for the Ar content, is located in accessible chromatin and CTCF regions [39], integrated with aseQTLs [9] and was previously identified as a QTL for REA [15].

DASE SNPs of DEGs overlapping FAANG’s epigenetic data are also integrated with potentially regulatory SNPs (Appendix A). The main SNPs in this context are: five variants overlapping meat-quality QTLs, five overlapping aseQTLs and seven with *cis*-eQTLs, as described in Table 2.

## 4. Discussion

The information on correlated phenotypes or the traits belonging to the same cluster is important to investigate the pleiotropic effect of allelic imbalance and understand DASE SNPs eventually identified for multiple meat quality traits. By correlating all phenotypes in our population, we found phenotypic interactions making biological sense. The Ca, Mg, and Zn minerals showed a positive correlation here and were associated with human muscle mass [41]. Zinc is an essential mineral for lean body mass synthesis, but relatively large amounts of Zn are needed for new tissue synthesis [42]. Moreover, we found positive correlations between unsaturated fatty acids and Cu and Cr. Chromium methionine chelate supplementation in Holstein steers increased the C20:4 (*p*-value = 0.07) and PUFA (*p*-value = 0.04) in beef during late fattening period [43].

Negative correlations were found between saturated fatty acid (SFA) composition and WBSF traits, which means that tender samples had larger concentrations of SFA. SFA can be related to initial and sustained juiciness in beef [44]. Although the excessive ingestion of SFA can be detrimental to the consumer’s health, they are still related to tenderness and juiciness in the bovine muscle [44,45]. To bring a balance to the production aspects, selecting a fatty acid composition in the herd must help the beef quality without increasing undesired fatty acids, which harm the consumers’ health.

The selection of contrasting samples for DASE analysis used almost all initially available animals (187 samples out of 190). Nevertheless, the DASE analysis was still conservative due to the limited number of animals in each contrasting group presenting collected phenotypes and a transcribed heterozygous SNP. The number of animals tested for DASE ranged from three to 18 per contrasting group. The statistical power of DASE analyses would be improved in larger experimental populations, in which it is possible to test larger contrasting groups because of the increased probability of heterozygous SNPs able to be tested. However, requiring at least three animals in each group may have guaranteed confidence in the obtained results without losing too much information due to our sample size.

Comparing the number of significant results with the ones from our previous work [9], we found that only 2.45% of the ASE SNPs were identified here as DASE SNPs. The proportion of tested SNPs in DASE analysis was 1.06% of the SNPs from the original dataset (~4 M SNPs), compared with the 3.02% tested for ASE [9]. Firstly, we did not expect all ASE SNPs to be classified as DASE SNPs because it is improbable that they all were located in genes related to the studied phenotypes. Moreover, the ASE results are obtained per animal, with multiple ASE SNPs being significant in a small proportion of the population, thus resulting in a larger number of significant results compared to the DASE analysis. Our analysis depends on the phenotypic variation, so it was performed in a selected sample rather than the whole population, resulting in a reduced number of significant results. The annotation of DASE and ASE SNPs was similar, mostly located in synonymous variation, downstream, intronic, and 3′ UTR regions.

DASE genes identified as hubs of the co-expression analysis were underlying metabolic pathways. ORA enriched for the protein ubiquitination process in M2. The ubiquitination process marks the proteins to be degraded in the cell cycle, requiring some calcium-dependent enzymes [46]. C18:2 increases ubiquitination and has a potential role in the proteasomal degradation of tyrosinase [47]. We found 15 DASE genes associated to C18:2 traits co-expressed in M2, being seven related to C18:2 *cis*-9, trans-11 (*ACIN1*, *COIL*, *DGKD*, *MED23*, *SMARCA4*, *TMEM167A*, and *RNF20*), five DASE genes for C18:2 trans-11, *cis*-15 (*ITGB1*, *RGSS*, *SCAP*, *BMS1*, and *CCNL2*) and three for C18:2 *cis*-9,12 trait (*EXOC1*, *NUTF2* and *TMX3*). From pig muscle, polymorphisms of ubiquitin protein ligase E3C were associated with IMF and fatty acid composition, being known as the relationship of E3 ubiquitin proteins and lipid metabolism [48]. Beef tenderness was related to proteassome proteolysis, which in turn, requires protein ubiquitination [49]. These results suggest an interplay between beef tenderness and fatty acid content in muscle, especially for C18:2, via ubiquitination-dependent proteasomal degradation.

To evaluate the enrichment of genes belonging to modules between the sample groups, the GSEA was performed. We found M1 and M2 modules with opposite effects in contrasting groups of the WBSF0 trait. The M1 module showed positive activity in the *Low WBSF0* group (more tender samples). Cytochrome b-c1 complex subunit 10 (*UQCR11),* a hub gene of M1, codes to the third complex of the mitochondria’s electron transport chain, and showed DASE for SFA, C18:3n6, C20:5, and PUFA. Fatty acids are related to mitochondrial membrane permeability [50]. After slaughter, under no blood circulation, mitochondria activity is impaired but still may influence biochemical processes in post-mortem muscle, which in turn requires oxygen from myoglobin to produce ATP [51]. *UCQCR11* converts ubiquinol in cytochrome c [52]. The release of cytochrome c induces apoptosis [51,53]. Apoptosis is a proteolytic process related to meat tenderization under oxidative stress [49,54,55], which can result from increased mitochondrial membrane permeability [50]. Utrophin, encoded by UTRN, the hub gene of M1, is a therapeutic target for Duchenne muscular disease [56,57]. The upregulation of this protein reduces oxidative stress and mitochondrial pathology [57], and this protein was related to the homeostasis of the sodium channel in the heart [56], attenuating the Duchenne’s pathology [56]. Among the 15 genes within M1 that showed DASE for the same enriched trait (WBSF0), there are Platelet-Derived Growth Factor Receptor Alpha (*PDGFRA)*, previously identified as a DEG for Ca [22], and Prune Homolog 2 With BCH Domain (*PRUNE2)*, differentially expressed in extremes of RFI [24].

The M2 module showed positive activity in the *High WBSF0* group (less tender samples), a reverse pattern compared with the M1 module. *SMAD5* is a hub gene of the M2 module, involved in 39 BPs (see Appendix A) being its product a transcription factor part of BMP signaling [58], exhibiting a role in myogenesis and muscle growth [58,59]. Oxidation resistance 1 (*OXR1*) was associated with growth traits in Wagyu cattle [60] and was found as a hub gene in our co-expression analysis. OXR1 has emerged as an essential antioxidant protein that controls neurons’ susceptibility to oxidative stress. The OXR1 protein is involved in maintaining mitochondrial morphology in the stress response as part of its antioxidant function [61]. This protein has orthologs between several species and is conserved throughout evolution [62]. Thus, the gene set of M1 seems to be related to meat tenderness, with positive activity in the tender group and the gene set of M2 have a role in muscle growth. These genes are probable candidates for improving beef quality and production, and the causal regulatory variants can be linked to the respective DASE SNPs.

Some DASE genes were associated with the same phenotypes affected by differentially expressed genes previously identified in the same population. These findings suggest that the total expression of these DASE/DE genes can be affected by *cis*-regulatory mechanisms acting over the given DASE SNP region. We identified three genes, SAM And SH3 Domain Containing 1 (*SASH1)*, Solute Carrier Family 25 Member 4 (*SLC25A4),* and Diacylglycerol Kinase Delta (*DGKD)* associated here and previously DEG with oleic acid content (C18:1 cis9) [19]. *SLC25A4* is a DEG up-regulated in the Low group for oleic acid content, showing ASE in the Low group in this work for its unique DASE SNP—rs109988743, located in a region with H3K4me3, H3K4me1, H3K27ac, and H3K27me3 peaks reported previously in bovine muscle [39]. *KLHL40* was found as DEG up-regulated in the Low group for palmitic acid (C16:0) [19], and its rs515929286 DASE SNP was found here with more ASE in the Low group, being located in an accessible chromatin region [39]. *STRIP2* was down-regulated in the Low group of REA [21] and showed ASE in the Low group of the same trait here for the rs137477165 DASE SNP, located in a functional region of H3K27ac histone modification [39]. The DASE SNPs pointed out here may be candidates to improve oleic acid content and REA in bovine muscle, and the variants regulating their transcription warrant further investigation.

The integration of DASE results and regulatory data revealed multiple functional variants overlapping or neighboring DASE SNPs, which might be responsible for the allelic expression pattern and phenotype. This study indicated that the DASE SNPs may not be exclusively markers for conditional-dependent ASE but also regulatory elements or in linkage disequilibrium with the causal variants. We showed DASE SNPs classified previously as *cis*-eQTLs and aseQTLs, such as rs519474617 and rs717173356, located in accessible chromatin regions and CTCF-binding regions, indicating CTCF-mediated transcription regulation [63]. DASE SNPs of previously described DEGs also integrated with multiple targets from FAANG’s study [39] and with aseQTLs, *cis*-eQTLs, and QTLs obtained in the same population, such as rs1117068355 and rs1117381943 DASE SNPs/aseQTLs, rs468833876 (DASE SNP/*cis*-eQTL) and rs457578905 (DASE SNP/QTL). Thus, DASE regions are co-localized with functional SNPs, which are useful for discovering regulatory variants that affect gene expression and phenotypes. Other studies can take advantage of the DASE results, highlighting which contrasting group presented ASE for the intended phenotype to track regulatory variants close to the DASE SNP or even admitting its own regulatory potential.

The DASE analysis is a helpful one-step approach for identifying genomic regions simultaneously associated with gene expression and phenotypic variance, as long as a large experimental population is available. From the DASE analysis, multiple potentially functional variants were identified located in previously reported regulatory regions. Because of the predicted association between DASE genes and meat quality traits, the identified DASE SNPs can be further explored to be applied in beef improvement programs.

## Figures and Tables

**Figure 1 genes-13-02336-f001:**
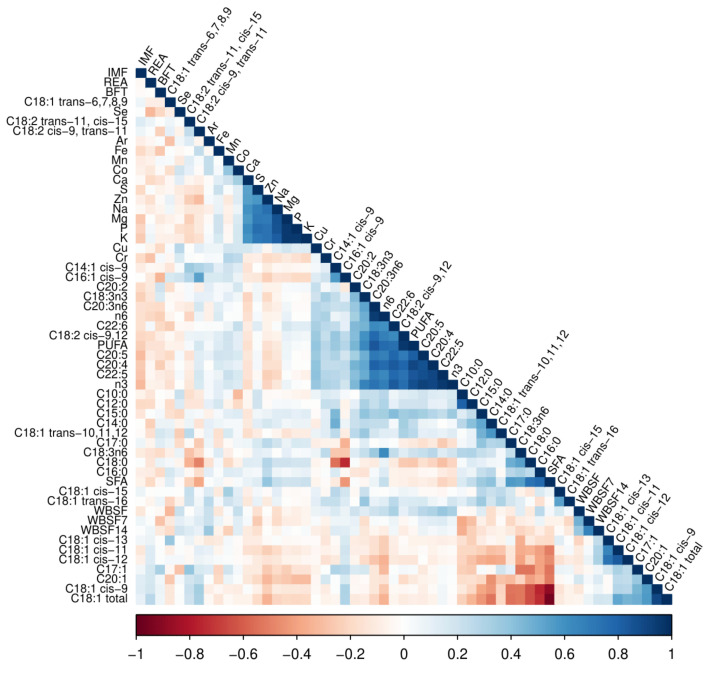
Correlogram of mineral and fatty acid content phenotypes, carcass and meat quality traits. Red squares represent negative correlations, blue squares represent positive correlations and the white squares represent phenotypes without correlation in this population.

**Figure 2 genes-13-02336-f002:**
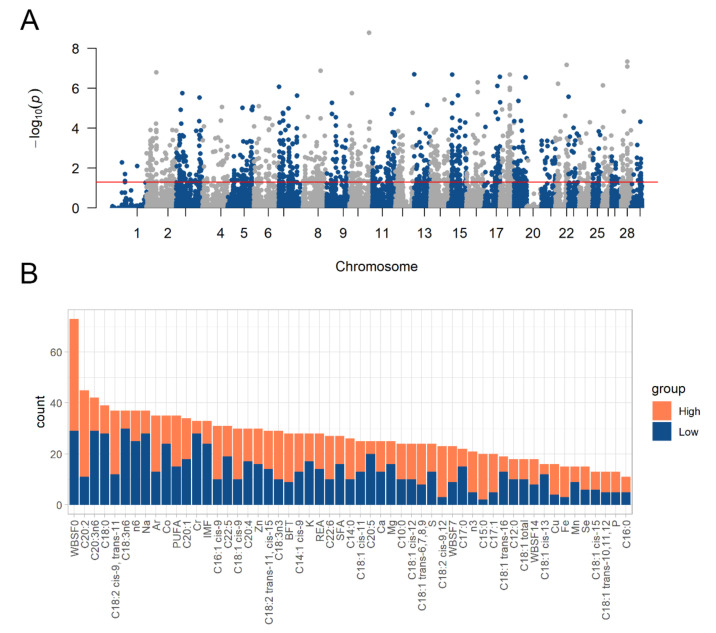
Differential allele-specific expression analysis results. (**A**) Manhattan Plot with all DASE analysis results. The points above the red line were the significant DASE SNPs (FDR ≤ 0.05). (**B**) Number of DASE SNPs per trait. The colors correspond to the sample group that presented more allelic imbalance: Orange color represent the number of DASE SNPs with more ASE in the high group (positive Log2FoldChange values), and the blue color represents the ones with more ASE in the low group (negative Log2FoldChange values).

**Figure 3 genes-13-02336-f003:**
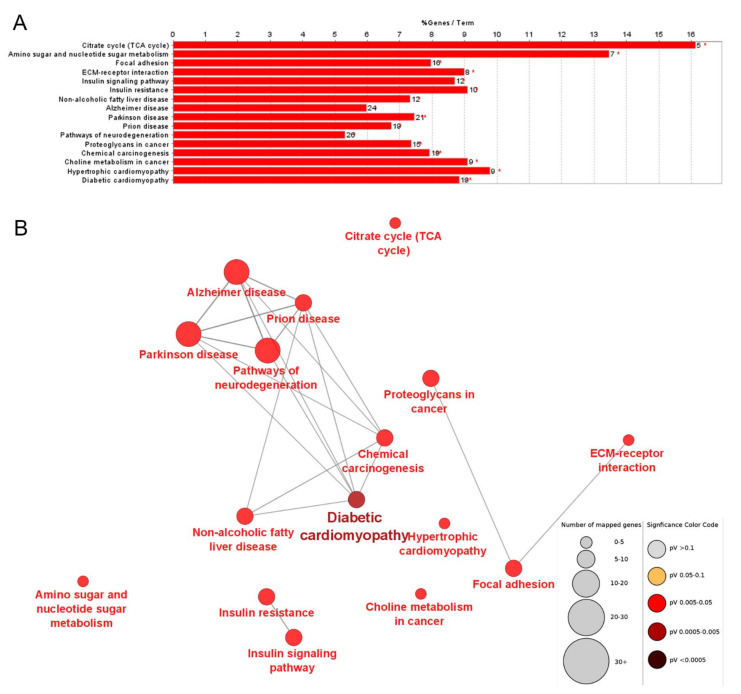
KEGG metabolic pathways enriched in the DASE genes. (**A**) Enrichment of KEGG biological pathways showing the number and proportion of DASE genes in the given pathways. (**B**) Interaction network of the enriched KEGG pathways.

**Figure 4 genes-13-02336-f004:**
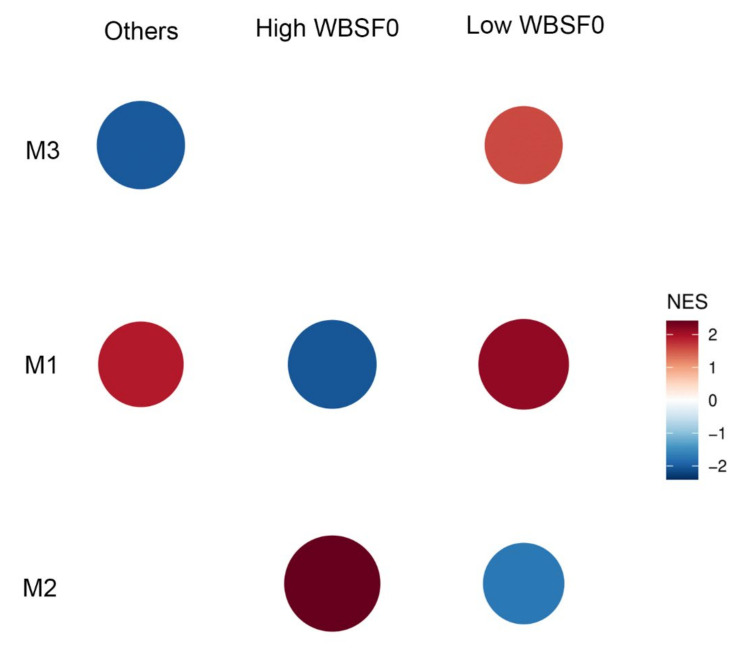
Gene set enrichment analysis of co-expression modules involving DASE genes for WBSF0 phenotype.

**Table 1 genes-13-02336-t001:** DASE SNPs overlapping QTLs associated with the same phenotypes in both analyses.

DASE SNP	Trait (DASE Analysis)	Trait (QTL Integration)
rs720456892, rs43664623, chr11:63468724 G>A, chr11:69714613 G>A	C18:0	C18:0
rs716062365, rs41721088, rs109947761	REA	REA
rs109976566, rs444293703	WBSF7	WBSF7
rs208417619, rs720858445	WBSF0	WBSF7

**Table 2 genes-13-02336-t002:** DASE SNPs located in DEGs which overlapped epigenetic data and putative regulatory variants.

DASE SNP	Variant Annotation/ Gene	Trait (DASE Analysis)	Trait (DE)	FAANG	Trait (QTL)	aseQTL	*cis*-eQTL
rs457578905	synonymous_variant/*THBS4*	Cr	P, Cu, Na, K, Mg, Average daily gain	ATAC-seq	LFAT	No	no
rs718541482	3_prime_UTR_variant/*WDR48*	PUFA	CLA-c9t11, PA	ATAC-seq	REA	No	no
rs211442363	3_prime_UTR_variant/*SPARC*	C20:1	IMF	H3K27ac	REA	No	no
rs715527852	downstream_gene_variant/*SUGT1*	P, S	CLA-c9t11	ATAC-seq	REA	No	no
rs1115255230	synonymous_variant/*SPARC*	n6, C20:1, C10:0	IMF	H3K27ac	REA	No	no
rs209388096	synonymous_variant/*AMFR*	C22:6	REA	ATAC-seq	no	Yes	no
rs719946630	synonymous_variant/*PGD*	WBSF14	CLA-c9t11	ATAC-seq	no	Yes	no
rs1117381943	upstream_gene_variant/*CAVIN2*	C18_1_T6_T7_T8_T9	AO	H3K4me3, H3K27ac, H3K4me1, H3K4me3, ATAC-seq, CTCF	no	Yes	no
rs516592412	synonymous_variant/*PDG*	C20:1	CLA-c9t11	ATAC-seq	no	Yes	no
rs1117068355	upstream_gene_variant/*CAVIN2*	C18_1_T6_T7_T8_T9	AO	H3K4me3, H3K4me1, H3K27me3, H3K27ac, ATAC-seq, CTCF	no	Yes	no
rs109090536	5_prime_UTR_variant/*DCAF11*	C20:2	AO	H3K4me3, H3K4me1, H3K27ac	no	No	yes
rs443738741	synonymous_variant/*LNX2*	C20:2	REA	ATAC-seq	no	No	yes
rs379719524	synonymous_variant/*NRAP*	Cr	AO	ATAC-seq	no	No	yes
rs715652252	synonymous_variant/*LNX2*	C20:2	REA	ATAC-seq, CTCF	no	No	yes
rs468833876	upstream_gene_variant/*NUTF2*	C18_2_C9_C12	CLA-c9t11	H3K27ac, H3K27me3, H3K4me3	no	No	yes
rs135451771	synonymous_variant/*NRAP*	Cr	AO	ATAC-seq	no	No	yes
chr2:5447956 G>A	synonymous_variant/*BIN1*	C18_1_T6_T7_T8_T9	AO	H3K27ac	no	No	yes

## Data Availability

The sequencing was archived on the European Nucleotide Archive (ENA) under accessions: PRJEB13188, PRJEB10898, and PRJEB19421.

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
