# Peer review of "Differential Allele-Specific Expression Revealed Functional Variants and Candidate Genes Related to Meat Quality Traits in B. indicus Muscle"

_genes, 2022, doi:10.3390/genes13122336_

Round 1

Reviewer 1 Report

The overall quality of the manuscript is high. However, some remarks might be mentioned. The authors should explain all the abbreviations when they first appear in the text. Usually the papers are aimed at scientific audience in the area, but they should also be easily understood and readable for scholars from other areas. For instance, please provide the full for SNP, and QTL. The material and method section is very detailed. However, it remains unclear if the authors use data from previous experiment, i.e. if the animals used are previously slaughtered? Also the produced steers are 386, however they use data from 190. the reason for this and how the animals were selected should be explained. Furthermore, section 2.1 should contain also description of the sampling procedure. However only muscle samples are considered here, while further blood and semen samples are also mentioned. 

The results of the work are clearly presented. The number of figures is adequate. The conclusions  are supported by the data. The level of English language is high.

Author Response

We appreciate your willingness to return our paper for revisions and further consideration for publication in the Genes journal. In the current version of the manuscript, we described the abbreviations, clarified some issues raised by the reviewer, and made minor corrections. Within the manuscript, the yellow highlights are related to the corrections.

Point 1: The authors should explain all the abbreviations when they first appear in the text.

Response 1: We reviewed the article, and all abbreviations were described in full the first time they were cited.

Point 2: The material and method section is very detailed. However, it remains unclear if the authors use data from previous experiment, i.e. if the animals used are previously slaughtered?

Response 2: The animals were part of a larger experimental population developed before this work and were already used in previous publications. In session 2.1, the following sentence was added:

“Details regarding the animal's production can be found in Tizioto et al. (2012, 2013) and Cesar et al. (2014).”

Point 3: Also the produced steers are 386, however they use data from 190. the reason for this and how the animals were selected should be explained. Furthermore, section 2.1 should contain also description of the sampling procedure.

Response 3: In this study, we used the data of 190 steers from the experimental population for which all phenotype, genotype, and RNA-Seq data were available. In addition, to ensure the comparability of the results, we used the same samples (N = 190) used in previous studies of our group (de Souza et al., 2020; Bruscadin et al., 2022). This information was inserted in the text. Please see session 2.1

Point 4: Only muscle samples are considered here, while further blood and semen samples are also mentioned.

Response 4: Semen and blood samples (used for genotyping and imputation) and muscle (used for RNA-seq) were obtained from the experimental population. This information was inserted in section  2.1. Animal production and sample collection

Reviewer 2 Report

This is an original work focused on the use of differential allele-specific expression (DASE) to identify regions in which allele-specific expression was confirmed in the phenotypic data of the meat quality trait under study. The present study used sophisticated techniques to identify specific genes and SNP polymorphisms that cause differential expression of alleles of functional genes and can be used in beef cattle breeding programs and also in understanding the heterosis effect in crossbred beef cattle. The results of the obtained research were properly described and discussed and brought some new knowledge.

Author Response

We appreciate your comments and consideration for publication in the Genes journal of our paper.